# PSEUDO MULTI-SOURCE DOMAIN GENERALIZATION: BRIDGING THE GAP BETWEEN SINGLE AND MULTI-SOURCE DOMAIN GENERALIZATION

## ABSTRACT

Deep learning models often struggle to maintain performance when deployed on data distributions different from their training data, particularly in real-world applications where environmental conditions frequently change. While multi-source domain generalization (MDG) has shown promise in addressing this challenge by leveraging multiple source domains during training, its practical application is limited by the significant costs and difficulties associated with creating multi-domain datasets. To address this limitation, we propose pseudo multi-source domain generalization (PMDG), a novel framework that enables the application of sophisticated MDG algorithms in a more practical single-source domain generalization setting. PMDG generates multiple pseudo-domains from a single source domain through style transfer and data augmentation techniques, creating a synthetic multi-domain dataset that can be used with MDG algorithms. Through extensive experiments with PseudoDomainBed, our modified version of the DomainBed benchmark, we analyze the effectiveness of PMDG across multiple datasets and architectures. Our analysis reveals several key findings, including a positive correlation between MDG and PMDG performance and the potential of pseudo-domains to match or exceed actual multi-domain performance with sufficient data. These comprehensive empirical results provide valuable insights for future research in domain generalization.

## 1 INTRODUCTION

Deep learning models have achieved remarkable success across various domains. However, their performance often deteriorates significantly when tested on data distributions different from their training data. This challenge is particularly prevalent in outdoor applications such as autonomous driving and smart cities, where environmental factors like weather conditions and lighting variations can substantially alter the input distribution. Therefore, developing robust deep learning models that maintain their performance on unseen distributions is crucial for real-world applications.

Domain generalization (DG) has emerged as a promising approach to address this challenge. DG techniques can be broadly categorized into two settings: single-source domain generalization (SDG), which uses data from a single source domain, and multi-source domain generalization (MDG), which leverages data from multiple source domains. While most existing research focuses on the MDG setting using multi-domain datasets (*e.g.*, PACS dataset (Li et al., 2017) with Photo, Art, Cartoon, and Sketch domains), creating such datasets is often impractical due to high collection and annotation costs. This limitation significantly hinders the practical application of MDG algorithms.

To bridge this gap, we propose pseudo multi-source domain generalization (PMDG), a novel framework that enables the application of sophisticated MDG algorithms in a more practical SDG setting. Our approach generates multiple pseudo-domains from a single source domain, treating them as distinct domains to create a synthetic multi-domain dataset. We investigate two approaches for effective pseudo-domain generation, style transformation, and data augmentation. For style transformation, inspired by the PACS dataset, we employ AdaIN Style Transfer (Huang & Belongie, 2017; Geirhos et al., 2018), CartoonGAN (Chen et al., 2018), and Edge Detection (Soria et al., 2020; 2023) to gen-

erate Art-style, Cartoon-style, and Sketch-style images. For data augmentation, we utilize various augmentation methods. An overview of PMDG is illustrated in Figure 1.

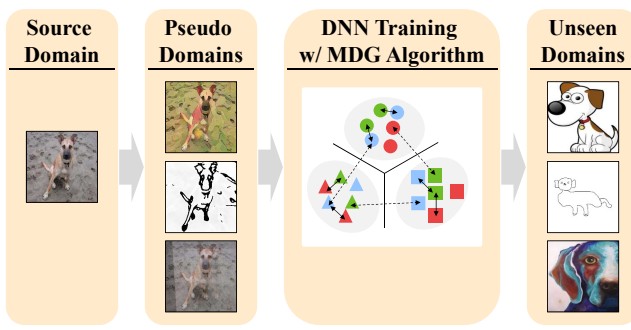

Figure 1: Overview of PMDG framework. PMDG applies multiple transformations to training samples to generate pseudo-domains. The DNN is then trained using an MDG algorithm on these pseudo-domains to be robust against unknown domains.

To evaluate PMDG, we create PseudoDomainBed, a modified version of the popular MDG benchmark DomainBed (Gulrajani & Lopez-Paz, 2020), adapted for the SDG setting. Through extensive experiments on multiple datasets and architectures (ResNet50 (He et al., 2016) and ViT (Dosovitskiy et al., 2021; Touvron et al., 2021)), we demonstrate that PMDG achieves superior performance compared to existing SDG baselines. Our analysis reveals key findings, including a positive correlation between MDG and PMDG performance, the potential of pseudo-domains to match or exceed actual multi-domain performance, and architecture-specific insights for pseudo-domain generation.

Our main contributions are as follows:

- We propose PMDG, a novel framework that bridges the gap between MDG and SDG, enabling sophisticated MDG algorithms in a practical SDG setting.
- We introduce PseudoDomainBed, a modified version of DomainBed with publicly available code. PseudoDomainBed facilitates easy utilization of MDG algorithms implemented in DomainBed and enables their evaluation in the SDG setting through pseudo-domains.
- Extensive experiments demonstrate that PMDG outperforms existing SDG methods and provide valuable insights to motivate future domain generalization research.

## 2 RELATED WORK

### 2.1 MULTI-SOURCE DOMAIN GENERALIZATION (MDG)

Most domain generalization research operates under the MDG paradigm, which assumes access to training data from multiple domains with shared label space. MDG approaches aim to learn domain-invariant information to improve generalization to unseen domains. Researchers have proposed various approaches, including learning domain-invariant features (Ganin et al., 2016; Li et al., 2018b; Sun & Saenko, 2016; Arjovsky et al., 2019; Motiian et al., 2017), regularization techniques (Sagawa et al., 2020; Huang et al., 2020; Krueger et al., 2021; Kim et al., 2021; Zhang et al., 2021; Shi et al., 2022; Chen et al., 2023; Pezeshki et al., 2021; Cha et al., 2021), data augmentation (Yan et al., 2020; Nam et al., 2021; Huang et al., 2020; Carlucci et al., 2019), self-supervised learning (Carlucci et al., 2019; Li et al., 2021; Kim et al., 2021), causal perspectives (Arjovsky et al., 2019; Krueger et al., 2021), meta-learning (Li et al., 2018a), and architectural innovations (Li et al., 2023).

While these approaches have shown promising results, their practical applicability has been limited by the requirement of multi-domain training data. Our proposed PMDG framework addresses this limitation by enabling MDG algorithms to operate effectively in a practical SDG setting.

### 2.2 SINGLE-SOURCE DOMAIN GENERALIZATION (SDG)

SDG research focuses on achieving domain generalization using training data from a single domain, which better reflects real-world scenarios. This approach is particularly relevant given that many widely-used computer vision datasets (*e.g.*, ImageNet (Russakovsky et al., 2015)) consist of data from a single domain. Current SDG approaches can be broadly categorized into three groups: learning algorithms, domain expansion methods, and data augmentation techniques. Learning al-

gorithms (Huang et al., 2020; Nam et al., 2021; Pezeshki et al., 2021) aim to prevent overfitting to source domain-specific information by introducing additional training objectives. Domain expansion methods systematically generate novel domains through various approaches: domain generator (Wang et al., 2021b; Li et al., 2021), uncertainty-guided generation (Qiao & Peng, 2021), optimal transport (Zhou et al., 2020), and adversarial data augmentation (Qiao et al., 2020; Volpi et al., 2018; Zhao et al., 2020). Data augmentation methods (Hendrycks et al., 2020; Huang et al., 2023; Hendrycks et al., 2022; Wang et al., 2021a; Vaish et al., 2024; Xu et al., 2021; Choi et al., 2023; Na et al., 2021; Kang et al., 2022; Zhou et al., 2021) focus on increasing the diversity of training data through various transformations, often with specialized training procedures.

While domain expansion methods offer sophisticated domain generation techniques, they often require complex adversarial attacks or specialized architectures that can be unstable and computationally intensive. In contrast, data augmentation methods provide a more straightforward and efficient approach to creating pseudo-domains, thus better suited for benchmarking MDG algorithms.

We treat augmented data as samples from different domains and utilize data augmentation techniques for pseudo-domain generation. Since identifying effective transformations and their combinations for pseudo-domain generation is non-trivial, we conducted empirical studies to address these questions. Our results demonstrate that pseudo-domains can serve as a practical testbed for the rich collection of MDG algorithms, suggesting that future research efforts should focus on developing effective pseudo-domain generation strategies rather than new training algorithms.

## 3 SDG PROBLEM SETTING

Our research follows the SDG problem setting. In SDG, we aim to learn a model that can generalize to unknown target domains using only a single source domain. Let $D^S = \{(x_i, y_i)\}_{i=1}^n$ be a source domain dataset, where $x_i \in \mathcal{X}$ represents input data, $y_i \in \mathcal{Y}$ represents labels, and $(x_i, y_i)$ follows the source domain distribution $P^S(X, Y)$. We consider a set of unknown target domains $\mathcal{T} = \{T_1, T_2, ..., T_k\}$, where each target domain $T_j$ has a different distribution from the source domain:

$$P^{T_j}(X, Y) \neq P^S(X, Y). \tag{1}$$

The SDG objective is to learn $f_\theta : \mathcal{X} \to \mathcal{Y}$ minimizing expected risk across target domains:

$$f_\theta^* = \arg\min_{f_\theta} \mathbb{E}_{T_j \in \mathcal{T}}[\mathbb{E}_{(x,y) \sim P^{T_j}}[L(f_\theta(x), y)]], \tag{2}$$

where $L : \mathcal{Y} \times \mathcal{Y} \to \mathbb{R}$ is a loss function.

## 4 PROPOSED FRAMEWORK

We propose pseudo multi-source domain generalization (PMDG), a novel framework that enables the application of MDG algorithms to single-source datasets by generating pseudo multi-domain datasets through various transformations. Algorithm 1 details its implementation.

---

**Algorithm 1** Training a DNN with PMDG

---

1: **Input:** Training dataset $D^S = \{(x_i, y_i)\}_{i=1}^n$,
   Number of epochs $E$, Batch size $B$, Transformations $\mathcal{O} = \{O_1, O_2, \ldots, O_N\}$
2: **Initialize** model parameters $\theta$
3: **for** epoch $e = 1$ to $E$ **do**
4:     Shuffle the training dataset $D^S$
5:     **for** each mini-batch $\{(x_b, y_b)\}_{b=1}^B$ in $D^S$ **do**
6:         Generate pseudo-domains: $\{\tilde{x}_b^k = O_k(x_b)\}_{k=1}^K$ for each image in mini-batch
7:         Obtain predictions: $\{\hat{y}_b^k = f_\theta(\tilde{x}_b^k)\}_{k=1}^K$ for each pseudo-domains
8:         Compute MDG loss: $L^{\mathrm{MDG}}$ using algorithm-specific objectives
9:         Update model parameters $\theta \leftarrow \theta - \eta \nabla_\theta L^{\mathrm{MDG}}$
10:    **end for**
11: **end for**
12: **Output:** Trained model parameters $\theta$

---

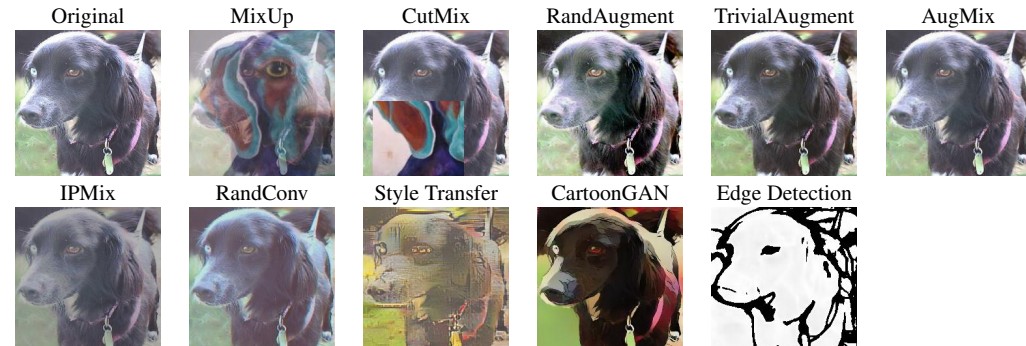

Figure 2: Visualization of transformations applied to dog images from the PACS dataset.

## 4.1 PSEUDO-DOMAIN GENERATION

Identifying effective methods for generating pseudo-domains remains a crucial open challenge. We introduce and evaluate two approaches, both independently and in combination, style transformation and data augmentation. Visual examples are presented in Figure 2.

**Style Transformation.** Inspired by the PACS dataset, we propose three transformations to recreate its constituent domains. The first transformation is AdaIN style transfer (Huang & Belongie, 2017; Geirhos et al., 2018), a technique that preserves image content (shape) while modifying style, used for creating art-style images. The second is CartoonGAN (Chen et al., 2018), a GAN-based approach for transforming images into cartoon-style representations. The third is Edge Detection (Soria et al., 2020; 2023), a method for extracting image contours used to generate sketch-style images.

**Data Augmentation.** We employ various data augmentation techniques, including mixing-based methods such as MixUp (Zhang et al., 2018) and CutMix (Yun et al., 2019), advanced augmentation strategies including RandAugment (Cubuk et al., 2020) and TrivialAugment (Müller & Hutter, 2021), and robustness-focused augmentations comprising AugMix (Hendrycks et al., 2020), IP-Mix (Huang et al., 2023), and RandConv (Xu et al., 2021). Although some augmentation techniques have associated loss functions, we omit them for simplicity in this study.

**Transformation Selection.** Given the limited understanding of optimal transformation count and inter-transformation interactions, we take an empirical approach. We select $K$ transformations (with replacement) from a predefined set of transformation operations to construct the transformation set $\mathcal{O} = \{O_1, O_2, \ldots, O_K\}$, where each $O$ represents an individual transformation operation.

**Transformation Application.** We generate pseudo multi-domain data using the transformation set $\mathcal{O}$. Specifically, for an input mini-batch $B = (x_i, y_i)_{i=1}^{b}$, we apply each transformation in the set to obtain $K$ pseudo multi-domain mini-batches $\{B_1, \ldots, B_K\}$:

$$B_k = O_k(B), \quad k = 1, \ldots, K. \tag{3}$$

## 4.2 TRAINING WITH MDG ALGORITHM

We train the model $f_\theta$ using an MDG algorithm on the $K$ pseudo multi-domain mini-batches, where our framework is algorithm-agnostic and can accommodate any MDG algorithm.

## 5 EXPERIMENTAL SETUP

We evaluated our approach using standard domain generalization datasets. Since model selection significantly impacts performance evaluation in domain generalization, we ensure fair comparison by modifying DomainBed (Gulrajani & Lopez-Paz, 2020), the standard MDG benchmark, to accommodate the SDG setting. We call our modified benchmark PseudoDomainBed, which implements our pseudo-domain generation approach. Implementation details are in the appendix.

## 5.1 IMPLEMENTATION DETAILS OF PSEUDODOMAINBED

Following the original implementation, we maintain consistent training configurations with DomainBed, including learning rate, batch size, and other hyperparameters. We use ResNet50 as our backbone network, maintaining the batch normalization layers (Ioffe & Szegedy, 2015) as per common practice in single-domain settings. For model selection, we employ training-domain validation sets, which has shown the most stable performance in MDG settings. Our pseudo-domain transformations are implemented as data augmentations applied at the mini-batch level, using default hyperparameters from their respective papers. These transformations are applied after the default data augmentation pipeline of DomainBed. Notably, while the original DomainBed applies data augmentation to validation sets, we omit this in PseudoDomainBed to avoid distorting the evaluation of generalization to unknown distributions.

Implementing various transformation techniques in a unified framework presents challenges due to their different operating levels. To address this, we carefully designed the implementation architecture of PseudoDomainBed to handle different types of transformations consistently. Image transformations in our framework can be categorized into two levels based on their processing stage. The first category is dataset-level transformations, which operate on raw images before converting them to tensors. Transformations such as RandAugment fall into this category, where augmentations are applied directly to image data. The second category is mini-batch-level transformations, which operate on normalized tensors during the training process. For example, MixUp belongs to this category as it combines multiple normalized image tensors. To handle these different transformation types uniformly, we implemented a standardized interface for pseudo-domain generation algorithms. Each algorithm is required to implement both dataset-level and mini-batch-level transformation methods, even if only one is actually used. This design choice provides a consistent API for users to employ any transformation technique without considering its implementation level. It also enables flexible integration of new transformation methods by implementing the standard interface.

## 6 RESULTS

We report the mean and standard error over three trials for each PseudoDomainBed experiment, demonstrating that PMDG outperforms SDG baselines.

### 6.1 EVALUATION OF PSEUDO-DOMAIN GENERATION

We first evaluated various transformation techniques for pseudo-domain generation in combination with different MDG algorithms on the VLCS dataset (Fang et al., 2013). In the experiment, we consider a two-domain setting consisting of the source domain and one pseudo-domain. To assess the effectiveness of each combination, we measure the accuracy gains from the ERM baseline (Guyon et al., 1991). Figure 3 shows a heatmap visualization of these results. IPMix, RandConv, and TrivialAugment show positive accuracy gains with most MDG algorithms, suggesting their effectiveness as pseudo-domain generation techniques. In contrast, CutMix leads to performance degradation in most cases. Notably, MLDG (Li et al., 2018a) shows performance deterioration across all transformations, suggesting its incompatibility with our pseudo-domain approach. The negative results with MLDG suggest that not all MDG algorithms are suitable for pseudo-domain settings, possibly due to their assumptions about domain characteristics.

### 6.2 ANALYSIS OF PSEUDO-DOMAIN COMBINATIONS

We evaluated two different pseudo-domain combinations with various MDG algorithms. The first combination consists of three domains, Org+IM+IM, while the second combination includes six domains, Org+ST+ED+CG+IM+IM. We compare these combinations using three learning algorithms: ERM, RIDG (Chen et al., 2023), and SD (Pezeshki et al., 2021). Table 1 presents the results. The combination of SD with Org+IM+IM achieves the highest accuracy of $55.9\%$, surpassing the best SDG baseline, IPMix ($55.2\%$). Interestingly, while the addition of style-based transformations (ST, ED, and CG) leads to significant improvements on the PACS dataset, its effectiveness is limited on other datasets. Evaluating methods solely on PACS might lead to the development of techniques that excel only on style-based domain shifts while failing to generalize to other types of

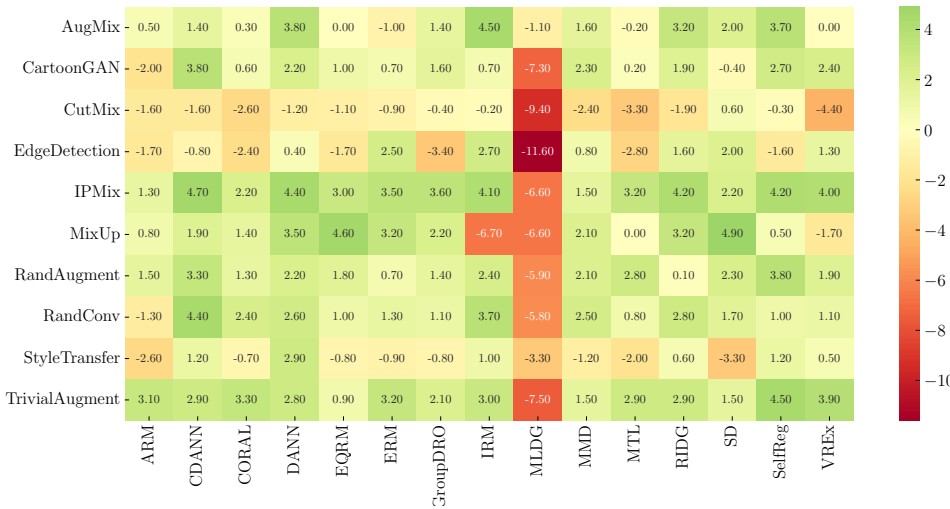

Figure 3: Accuracy gains over the ERM baseline without pseudo-domain across different transformation techniques (y-axis) and MDG algorithms (x-axis) on the VLCS dataset. Green and red colors indicate performance improvements and degradation, respectively. Values represent accuracy differences from ERM without pseudo-domains.

distribution shifts. This observation emphasizes the importance of diverse evaluation protocols, including multiple datasets with different domain shifts. While specialized techniques like style-based transformations are highly effective for specific scenarios, they should be complemented by more general-purpose approaches for broader applicability.

Additionally, to investigate the effectiveness of our PMDG framework across different architectures, we conducted experiments using Vision Transformer (ViT) as the backbone network. Table 1 presents the results. The combination of SD with pseudo-domains Org+ST+ED+CG+IM+IM achieves the highest accuracy of 62.0%, surpassing the best SDG baseline (60.5%). Style-based transformations that enhance shape features appear to be more effective with ViT (Li et al., 2020; Tuli et al., 2021), suggesting that the choice of pseudo-domain generation techniques should consider the underlying architectural characteristics of the backbone network.

## 6.3 CORRELATION WITH MDG PERFORMANCE

To understand which MDG algorithms are most suitable for PMDG, we investigated whether the optimal choice of MDG algorithms differs between MDG and PMDG settings, as the training domains in these settings are fundamentally different (real domains vs. pseudo-domains). For simplicity, we used Org+IM as our pseudo-domain configuration. Figure 4 visualizes the relationship between algorithm performance in MDG and PMDG settings. The results show a positive correlation between MDG and PMDG accuracy. This correlation has two important implications. First, it suggests that MDG algorithms that perform well in traditional multi-domain settings are also well-suited for our pseudo-domain approach. Second, it suggests that the PMDG framework could enhance the practical utility of MDG algorithms by enabling their application to single-domain problems.

## 6.4 QUALITY ASSESSMENT: PMDG VS. MDG

We conducted a comparison between PMDG and conventional MDG approaches under controlled data conditions to assess whether artificially generated pseudo-domains can serve as effective substitutes for naturally occurring domain variations. To ensure a fair comparison, we set the total number of training samples equal in both settings. Given $n$ samples from a single source domain in the PMDG setting, we constructed an MDG training set by collecting $n/3$ samples from each of three distinct domains, maintaining the same total size $n$. These source domain samples were transformed using our Org+IM+IM pseudo-domain generation method in the PMDG setting, while the MDG setting used the original samples directly. Both approaches used SD as the base algorithm

Table 1: Accuracy comparison on four datasets using different combinations of algorithms and pseudo-domains. Upper/lower tables show ResNet50/ViT results, respectively. Org denotes the original domain without transformation, IM represents pseudo-domains generated by IPMix, ST by StyleTransfer, ED by EdgeDetection, and CG by CartoonGAN. $^\dagger$ indicates exclusion of domain-specific transformations during training: ST is excluded when testing on Art domain, CG for Cartoon domain, and ED for Sketch domain. "Avg" represents the mean accuracy across all datasets. In each table, the upper part shows SDG baseline results and the lower part presents PMDG results. **Bold** and underlined numbers denote the first and second highest accuracy, respectively.

| Algorithm | Pseudo-domain | VLCS | PACS | OfficeHome | TerraIncognita | Avg |
|---|---|---|---|---|---|---|
| ERM | – | $60.6 \pm 1.3$ | $56.3 \pm 0.5$ | $53.4 \pm 0.1$ | $32.2 \pm 0.7$ | 50.6 |
| Mixup | – | $64.3 \pm 1.2$ | $55.9 \pm 0.8$ | $54.5 \pm 0.3$ | $32.7 \pm 0.5$ | 51.8 |
| SagNet | – | $62.0 \pm 0.3$ | $52.2 \pm 0.4$ | $51.5 \pm 0.3$ | $32.6 \pm 0.7$ | 49.6 |
| RSC | – | $\underline{65.0} \pm 0.9$ | $56.1 \pm 1.3$ | $52.5 \pm 0.3$ | $33.3 \pm 0.2$ | 51.7 |
| AugMix | – | $61.8 \pm 0.5$ | $57.8 \pm 0.4$ | $54.6 \pm 0.3$ | $32.1 \pm 0.3$ | 51.6 |
| CutMix | – | $61.8 \pm 0.3$ | $54.7 \pm 1.0$ | $53.9 \pm 0.1$ | $33.0 \pm 1.1$ | 50.8 |
| IPMix | – | $64.6 \pm 1.0$ | $65.9 \pm 0.3$ | $\underline{55.6} \pm 0.2$ | $34.9 \pm 0.7$ | 55.2 |
| RandAugment | – | $58.6 \pm 0.8$ | $58.9 \pm 1.0$ | $53.9 \pm 0.2$ | $33.2 \pm 0.5$ | 51.1 |
| RandConv | – | $62.1 \pm 0.1$ | $62.8 \pm 0.7$ | $53.2 \pm 0.3$ | $34.7 \pm 0.3$ | 53.2 |
| TrivialAugment | – | $61.1 \pm 1.1$ | $59.9 \pm 1.5$ | $54.2 \pm 0.2$ | $36.2 \pm 0.2$ | 52.8 |
| ERM | Org+IM+IM | $64.6 \pm 1.5$ | $63.4 \pm 1.3$ | $55.1 \pm 0.4$ | $\underline{36.6} \pm 0.9$ | 54.9 |
| ERM | Org+ST+ED+CG+IM+IM$^\dagger$ | $64.9 \pm 1.1$ | $69.9 \pm 0.5$ | $55.4 \pm 0.1$ | $31.1 \pm 0.6$ | $\underline{55.3}$ |
| RIDG | Org+IM+IM | $63.4 \pm 1.6$ | $64.8 \pm 0.3$ | $55.4 \pm 0.5$ | $37.2 \pm 0.2$ | 55.2 |
| RIDG | Org+ST+ED+CG+IM+IM$^\dagger$ | $61.7 \pm 0.0$ | $\mathbf{71.8} \pm 0.4$ | $55.2 \pm 0.2$ | $31.7 \pm 0.4$ | 55.1 |
| SD | Org+IM+IM | $\mathbf{65.6} \pm 1.2$ | $64.1 \pm 1.0$ | $\mathbf{56.7} \pm 0.2$ | $\mathbf{37.1} \pm 0.7$ | $\mathbf{55.9}$ |
| SD | Org+ST+ED+CG+IM+IM$^\dagger$ | $61.4 \pm 0.3$ | $\underline{69.7} \pm 0.4$ | $\underline{55.6} \pm 0.2$ | $30.5 \pm 0.9$ | 54.3 |

| Algorithm | Pseudo-domain | VLCS | PACS | OfficeHome | TerraIncognita | Avg |
|---|---|---|---|---|---|---|
| ERM | – | $64.5 \pm 0.8$ | $73.7 \pm 0.5$ | $66.7 \pm 0.4$ | $32.1 \pm 0.8$ | 59.3 |
| IPMix | – | $65.8 \pm 1.0$ | $76.3 \pm 1.0$ | $66.2 \pm 0.3$ | $33.9 \pm 0.1$ | 60.5 |
| ERM | Org+IM+IM | $\mathbf{67.8} \pm 0.1$ | $74.5 \pm 1.2$ | $67.3 \pm 0.2$ | $33.7 \pm 0.2$ | 60.8 |
| ERM | Org+ST+ED+CG+IM+IM$^\dagger$ | $66.4 \pm 0.6$ | $79.6 \pm 0.3$ | $66.8 \pm 0.1$ | $31.3 \pm 0.4$ | 61.0 |
| RIDG | Org+IM+IM | $\underline{67.6} \pm 0.3$ | $76.1 \pm 0.8$ | $68.1 \pm 0.4$ | $\mathbf{35.7} \pm 0.7$ | $\underline{61.9}$ |
| RIDG | Org+ST+ED+CG+IM+IM$^\dagger$ | $64.4 \pm 0.3$ | $\underline{80.8} \pm 0.3$ | $67.9 \pm 0.2$ | $32.0 \pm 0.5$ | 61.3 |
| SD | Org+IM+IM | $67.5 \pm 1.8$ | $\underline{76.7} \pm 0.6$ | $\underline{68.4} \pm 0.1$ | $\underline{34.3} \pm 0.3$ | 61.7 |
| SD | Org+ST+ED+CG+IM+IM$^\dagger$ | $66.4 \pm 0.7$ | $\mathbf{81.3} \pm 0.3$ | $\mathbf{68.6} \pm 0.1$ | $31.7 \pm 0.8$ | $\mathbf{62.0}$ |

for domain generalization. The experimental results shown in Figure 5 reveal several intriguing patterns. Overall, MDG with actual multi-domain data demonstrates superior performance compared to PMDG. However, this performance gap narrows as the number of training samples increases. More notably, when evaluating on specific test domains (C and S), PMDG actually outperforms MDG in scenarios with larger training datasets. This performance inversion suggests that pseudo-domain generation becomes increasingly effective with more training data, potentially due to the model's enhanced ability to learn meaningful domain transformations from a richer source dataset. These findings have significant implications for domain generalization research. The convergence and occasional superiority of PMDG performance with larger datasets indicate that synthetic domain generation could serve as a viable, and sometimes preferable, alternative to costly multi-domain data collection. This is particularly relevant for scenarios where collecting large-scale multi-domain data is impractical or resource-intensive. Moreover, the domain-specific nature of PMDG's advantages suggests that our artificial domain transformations might capture certain aspects of domain variation, particularly well for specific target domains. This finding implies that with further refinement of pseudo-domain generation techniques and sufficient training data, PMDG could offer a more scalable and cost-effective approach to domain generalization while maintaining or even exceeding the performance of traditional MDG methods.

## 6.5 EVALUATION ON IMAGENET

In our previous experiments, we established the effectiveness of combining MDG algorithms with pseudo-domains. We then investigated whether this insight could benefit existing SDG research on

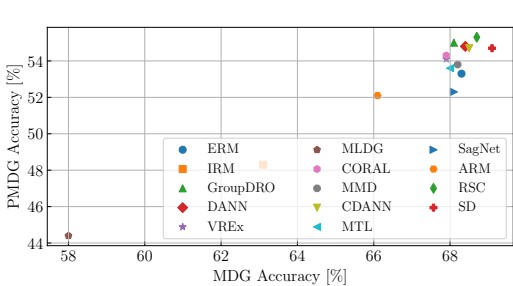

Figure 4: Accuracy comparison of MDG algorithms across MDG and PMDG settings. Each point represents a different MDG algorithm. Accuracy represents averages across PACS, VLCS, OfficeHome (Venkateswara et al., 2017), and TerraIncognita (Beery et al., 2018) datasets.

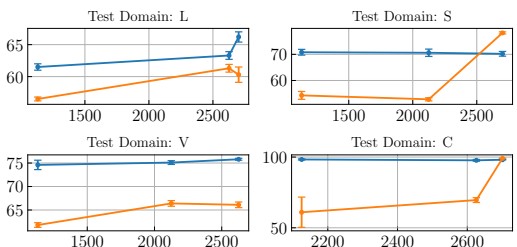

Figure 5: Comparison of accuracy between MDG and PMDG settings on the VLCS dataset under equal training data conditions. The x-axis shows the total number of training samples, while the y-axis shows the accuracy. Blue and orange lines represent MDG and PMDG settings, respectively. Each subplot corresponds to a different test domain: LabelMe (top-left), VOC2007 (bottom-left), SUN09 (top-right), and Caltech101 (bottom-right).

Table 2: Accuracy comparison of IPMix and our method on the ImageNet (IN) dataset and its variants. "Avg" represents the mean accuracy across all test datasets, and "OOD Avg" denotes the mean accuracy on out-of-distribution test datasets.

| Method | IN | IN-C | IN-R | IN-A | IN-$\overline{\text{C}}$ | IN-Sketch | IN-V2 | Stylized-IN | Avg | OOD Avg |
|--------|-----|------|------|------|------|-----------|-------|-------------|-----|---------|
| IPMix | **77.71** | 51.01 | 42.67 | 4.27 | **52.11** | 31.07 | 65.61 | 11.54 | 42.00 | 36.90 |
| Ours | 77.31 | **51.18** | **43.78** | **6.13** | 49.71 | **31.24** | **65.91** | **12.44** | **42.21** | **37.20** |

large-scale datasets. Specifically, we evaluated our approach on ImageNet, a standard benchmark in SDG research that offers various distribution shifts through its variants (Russakovsky et al., 2015; Hendrycks et al., 2021b; Hendrycks & Dietterich, 2019; Mintun et al., 2021; Hendrycks et al., 2021a; Wang et al., 2019; Recht et al., 2019; Geirhos et al., 2018). We utilized SD as the MDG algorithm with pseudo-domains Org+IM+IM, comparing it against the baseline SDG method IPMix. The selection of IPMix as our baseline is particularly relevant as it represents a state-of-the-art SDG approach. The results, presented in Table 2, demonstrate the broad applicability of our findings. Our method achieves better average accuracy across out-of-distribution variants than the IPMix baseline, showing improved generalization performance. These results offer several important implications for the SDG research community. First, they validate that insights from MDG can indeed enhance existing SDG methods, even on large-scale datasets. Second, the performance improvements across multiple distribution shifts suggest that our approach can effectively capture robust features. Finally, these findings open new possibilities for advancing SDG research by incorporating established MDG techniques, potentially bridging the gap between these separate research directions.

## 7 DISCUSSION

Based on our experimental results, we present several key insights that have important implications for future research in domain generalization. Through our findings, we aim to bridge the gap between SDG and MDG research.

**Incorporating MDG Advances into SDG Research.** We revealed a positive correlation between algorithm performance in MDG and PMDG settings. Furthermore, the combination of SD algorithm with pseudo-domains Org+IM+IM achieved the highest performance, surpassing SDG baselines. These results demonstrate that leveraging established MDG algorithms through our PMDG framework can enhance SDG performance. The successful application of MDG algorithms in the SDG setting suggests that the traditional separation between these fields may have unnecessarily limited SDG research progress. PMDG framework provides a practical testbed for MDG algorithms, bridging the gap between MDG advances and their practical application in SDG scenarios.

**Reconsidering the Role of SDG Research.** Our experimental results revealed two important findings. First, pseudo-domains can sometimes outperform actual multi-domains when sufficient training data is available, and MDG algorithms work effectively as learning algorithms in the SDG setting. These findings suggest that the quality of pseudo-domain generation may have a greater impact on generalization performance than the development of new learning algorithms, as existing MDG algorithms already provide strong learning capabilities. Furthermore, our comparison between MDG and PMDG under equal data conditions revealed that the performance gap between them diminishes as training data increases, with PMDG even showing superiority in some cases. This observation challenges the conventional assumption that real multi-domain data is always preferable and suggests that with sufficient data, well-designed pseudo-domain generation might be more effective than collecting actual multi-domain datasets. Consequently, we argue that future SDG research should prioritize the development of better pseudo-domain generation techniques rather than creating new learning algorithms in isolation from MDG advances. This focus shift could yield substantial performance improvements.

**Future Directions.** A key direction for future research is the theoretical analysis of when and why pseudo-domains can substitute for actual domains. This analysis could provide insights into the fundamental principles of domain generalization and guide the development of more effective pseudo-domain generation techniques. The success of our PMDG framework demonstrates that the artificial boundary between SDG and MDG research has been limiting progress in both fields. By bridging these traditionally separate areas, we suggest that future advances in domain generalization may come from their synergistic combination: utilizing sophisticated MDG algorithms while focusing SDG research efforts on improving pseudo-domain generation techniques.

## 8 LIMITATION

A key limitation of our current PMDG framework is its underlying assumption that all transformed data represents distributions distinct from the source data distribution. This limitation becomes particularly problematic when dealing with varying transformation intensities, as weakly transformed data may remain substantially similar to the source distribution. Consequently, treating such transformed data as distinct domains could lead to suboptimal performance by artificially emphasizing minor variations that do not contribute to meaningful domain generalization. While addressing this limitation would require extensive theoretical analysis and empirical validation, a potential solution would be to quantitatively assess the relationship between source and transformed distributions. Such analysis, although computationally intensive, could provide a more principled foundation for pseudo-domain generation and potentially lead to more effective domain generalization strategies.

## 9 CONCLUSION

We proposed the PMDG framework to bridge the gap between SDG and MDG research. Through extensive experiments, we demonstrated that incorporating MDG algorithms into the SDG setting via pseudo-domain generation can enhance generalization performance. We obtained several important findings from our experiments. First, MDG algorithms can be effectively utilized in the SDG setting through our PMDG framework, achieving state-of-the-art performance. Second, when sufficient training data are available, pseudo-domains can serve as effective substitutes for actual multi-source domains, suggesting that future SDG research should focus on developing better pseudo-domain generation techniques rather than new training algorithms. Third, the effectiveness of pseudo-domain generation techniques can vary with network architectures, as demonstrated by our experiments. Our work also reveals important considerations for domain generalization research. The dataset-specific effectiveness of certain pseudo-domain combinations (*e.g.*, style-based transformations for PACS) highlights the importance of evaluating methods across diverse distribution shifts. These findings open new directions for future research, particularly in understanding the theoretical relationship between pseudo-domains and actual domains. By providing a bridge between SDG and MDG research, our work suggests that future advances in domain generalization may come from their synergistic combination rather than treating them as separate fields.

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

In this Supplementary Material, we provide additional analysis for the PMDG.

## A  DETAILS OF EXPERIMENTAL SETUP

### A.1  DATASETS

Our primary evaluation uses four standard domain generalization datasets: PACS (Li et al., 2017), VLCS (Fang et al., 2013), OfficeHome (Venkateswara et al., 2017), and TerraIncognita (Beery et al., 2018).

### A.2  SDG BASELINES

We compare our method against standard SDG baselines: ERM (Guyon et al., 1991), Mixup (Zhang et al., 2018), SagNet (Nam et al., 2021), RSC (Huang et al., 2020), AugMix (Hendrycks et al., 2020), CutMix (Yun et al., 2019), IPMix (Huang et al., 2023), RandAugment (Cubuk et al., 2020), RandConv (Xu et al., 2021) and TrivialAugment (Müller & Hutter, 2021). These baselines represent the current state-of-the-art approaches in single-domain generalization.

### A.3  MDG ALGORITHMS

We utilize the MDG algorithms implemented in DomainBed with their default hyperparameters. Specifically, we evaluate our approach with algorithms ARM (Zhang et al., 2021), CDANN (Li et al., 2018c), CORAL (Sun & Saenko, 2016), DANN (Ganin et al., 2016), EQRM (Eastwood et al., 2022), GroupDRO (Sagawa et al., 2020), IRM (Arjovsky et al., 2019), MLDG (Li et al., 2018a), MMD (Li et al., 2018b), MTL (Blanchard et al., 2021), RIDG (Chen et al., 2023), SD (Pezeshki et al., 2021), SelfReg (Kim et al., 2021) and VREx (Krueger et al., 2021).

## B  EXTENDED ANALYSIS OF PSEUDO-DOMAIN COMBINATIONS

While the main paper focuses on two specific pseudo-domain combinations, here we present a comprehensive evaluation of various combinations to better understand their effectiveness. We evaluated these combinations using ERM and SD as the MDG algorithms, as they demonstrated strong performance in our main experiments. Table 3 presents the results of this extended analysis. The experimental results revealed several important insights about pseudo-domain combinations. First, not all combinations lead to improved performance, with some degrading accuracy compared to using no pseudo-domains. This negative effect is particularly pronounced when incorporating CutMix into the combinations, suggesting that certain transformations may interfere with the model's ability to learn robust features. Second, we found that applying IPMix multiple times consistently outperforms combinations of different data augmentation techniques. This suggests that the quality and consistency of the pseudo-domain generation technique may be more important than the diversity of transformation types. Third, while style-based transformations alone show moderate effectiveness, their combination with IPMix leads to further improvements in performance. This complementary effect indicates that style-based transformations and IPMix capture different aspects of domain variation, making their combination particularly effective for domain generalization.

## C  ADDITIONAL RESULTS OF QUALITY ASSESSMENT

While the main paper presents our data efficiency analysis on the VLCS dataset, we conducted similar experiments on PACS, OfficeHome, and TerraIncognita datasets. Following the same experimental setup, we evaluated both MDG and PMDG settings under equal training data conditions. Given $n$ samples from a single source domain in the PMDG setting, we constructed an MDG training set by collecting $n/3$ samples from each of three distinct domains, maintaining the same total size $n$. The PMDG setting used Org+IM+IM pseudo-domains, while the MDG setting used the original samples directly. Both approaches used SD as the MDG algorithm for domain generalization.

Figures 6, 7, and 8 show the results for the PACS, OfficeHome, and TerraIncognita datasets, respectively. The experimental results revealed that MDG performance generally improves proportionally with the increase in training data size across all datasets, suggesting that MDG approaches can effectively leverage larger amounts of diverse training data. Similar to our findings on the VLCS

Table 3: Accuracy comparison of different pseudo-domain combinations on four datasets. "Avg" represents the mean accuracy across all datasets. In each table, the upper part shows SDG baseline results and the lower part presents PMDG results. Org denotes the original domain without transformation, IM represents pseudo-domains generated by IPMix, AM by AugMix, MU by Mixup, CM by CutMix, RC by RandConv, ST by StyleTransfer, ED by EdgeDetection, and CG by CartoonGAN. † indicates exclusion of domain-specific transformations during training: ST is excluded when testing on Art domain, CG for Cartoon domain, and ED for Sketch domain.

| Algorithm | Pseudo-domain | VLCS | PACS | OfficeHome | TerraIncognita | Avg |
|---|---|---|---|---|---|---|
| ERM | – | $60.6 \pm 1.3$ | $56.3 \pm 0.5$ | $53.4 \pm 0.1$ | $32.2 \pm 0.7$ | 50.6 |
| Mixup | – | $64.3 \pm 1.2$ | $55.9 \pm 0.8$ | $54.5 \pm 0.3$ | $32.7 \pm 0.5$ | 51.8 |
| SagNet | – | $62.0 \pm 0.3$ | $52.2 \pm 0.4$ | $51.5 \pm 0.3$ | $32.6 \pm 0.7$ | 49.6 |
| RSC | – | $65.0 \pm 0.9$ | $56.1 \pm 1.3$ | $52.5 \pm 0.3$ | $33.3 \pm 0.2$ | 51.7 |
| AugMix | – | $61.8 \pm 0.5$ | $57.8 \pm 0.4$ | $54.6 \pm 0.3$ | $32.1 \pm 0.3$ | 51.6 |
| CutMix | – | $61.8 \pm 0.3$ | $54.7 \pm 1.0$ | $53.9 \pm 0.1$ | $33.0 \pm 1.1$ | 50.8 |
| IPMix | – | $64.6 \pm 1.0$ | $65.9 \pm 0.3$ | $55.6 \pm 0.2$ | $34.9 \pm 0.7$ | 55.2 |
| RandAugment | – | $58.6 \pm 0.8$ | $58.9 \pm 1.0$ | $53.9 \pm 0.2$ | $33.2 \pm 0.5$ | 51.1 |
| RandConv | – | $62.1 \pm 0.1$ | $62.8 \pm 0.7$ | $53.2 \pm 0.3$ | $34.7 \pm 0.3$ | 53.2 |
| TrivialAugment | – | $61.1 \pm 1.1$ | $59.9 \pm 1.5$ | $54.2 \pm 0.2$ | $36.2 \pm 0.2$ | 52.8 |
| ERM | Org+AM+MU+CM | $59.0 \pm 0.6$ | $51.6 \pm 0.6$ | $51.8 \pm 0.2$ | $34.9 \pm 0.2$ | 49.4 |
| ERM | Org+IM+IM | $64.6 \pm 1.5$ | $63.4 \pm 1.3$ | $55.1 \pm 0.4$ | $36.6 \pm 0.9$ | 54.9 |
| ERM | Org+IM+RC | $65.0 \pm 0.5$ | $61.3 \pm 0.6$ | $55.9 \pm 0.3$ | $33.8 \pm 0.9$ | 54.0 |
| ERM | Org+ST+ED+CT† | $60.2 \pm 0.7$ | 59.7 | 55.3 | $28.2 \pm 0.7$ | 50.9 |
| ERM | Org+ST+ED+CG+IM+IM† | $64.9 \pm 1.1$ | $69.9 \pm 0.5$ | $55.4 \pm 0.1$ | $31.1 \pm 0.6$ | 55.3 |
| SD | AM+IM | $63.8 \pm 0.6$ | $63.5 \pm 1.5$ | $55.5 \pm 0.3$ | $37.5 \pm 0.6$ | 55.1 |
| SD | MU+IM | $62.2 \pm 0.6$ | $54.1 \pm 1.0$ | $54.7 \pm 0.2$ | $36.8 \pm 0.7$ | 51.9 |
| SD | Org+AM+IM | $65.4 \pm 1.2$ | $62.1 \pm 0.5$ | $56.3 \pm 0.4$ | $37.0 \pm 0.2$ | 55.2 |
| SD | Org+AM+MU | $62.0 \pm 1.0$ | $57.1 \pm 0.9$ | $54.3 \pm 0.3$ | $37.3 \pm 1.1$ | 52.7 |
| SD | Org+AM+MU+CM | $60.9 \pm 0.9$ | $54.5 \pm 0.7$ | $52.4 \pm 0.4$ | $35.4 \pm 0.5$ | 50.8 |
| SD | Org+AM+MU+CM+IM | $62.0 \pm 0.3$ | $55.3 \pm 0.4$ | $53.8 \pm 0.9$ | $34.7 \pm 1.1$ | 51.5 |
| SD | Org+MU+IM | $65.1 \pm 1.1$ | $58.9 \pm 0.6$ | $55.0 \pm 0.2$ | $38.7 \pm 0.9$ | 54.4 |
| SD | Org+IM+RC | $63.7 \pm 0.7$ | $63.1 \pm 0.7$ | $56.6 \pm 0.2$ | $36.8 \pm 0.5$ | 55.1 |
| SD | Org+IM | $62.8 \pm 1.3$ | $62.3 \pm 0.5$ | $56.3 \pm 0.1$ | $37.2 \pm 1.3$ | 54.7 |
| SD | Org+IM+IM | $65.6 \pm 1.2$ | $64.1 \pm 1.0$ | $56.7 \pm 0.2$ | $37.1 \pm 0.7$ | 55.9 |
| SD | Org+IM+IM+IM | $66.0 \pm 1.1$ | $61.0 \pm 0.6$ | $56.8 \pm 0.1$ | $37.4 \pm 0.1$ | 55.3 |
| SD | Org+ST+ED+CT† | $61.3 \pm 1.0$ | 60.5 | 56.1 | $30.0 \pm 0.4$ | 52.0 |
| SD | Org+ST+ED+CG+IM+IM† | $61.4 \pm 0.3$ | $69.7 \pm 0.4$ | $55.6 \pm 0.2$ | $30.5 \pm 0.9$ | 54.3 |

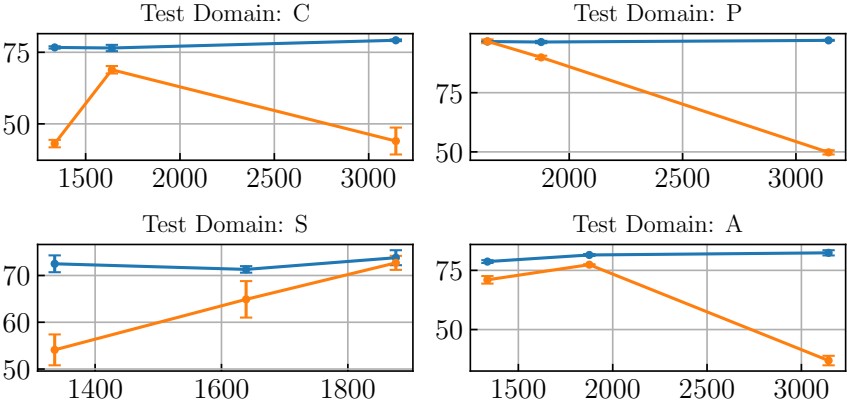

Figure 6: Comparison of accuracy between MDG and PMDG settings on the PACS dataset under equal training data conditions. The x-axis shows the total number of training samples, while the y-axis shows the accuracy. Blue and orange lines represent MDG and PMDG settings, respectively. Each subplot corresponds to a different test domain: Cartoon (top-left), Sketch (bottom-left), Photo (top-right), and Art Painting (bottom-right).

dataset, PMDG occasionally outperforms MDG under specific conditions. The effectiveness of PMDG varies significantly depending on both source and test domain combinations.

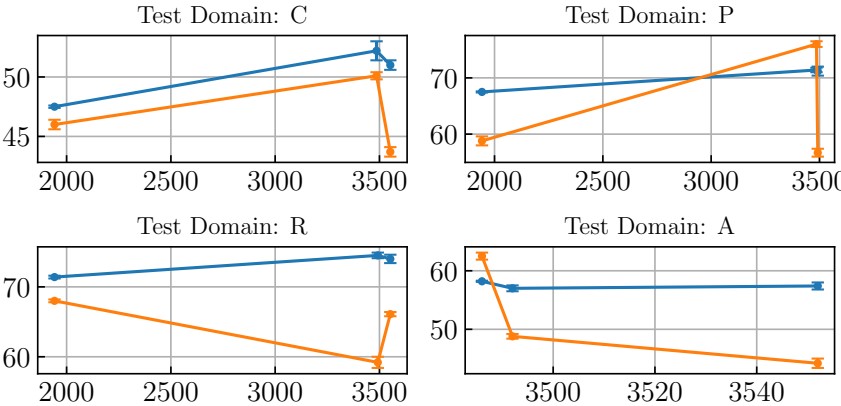

Figure 7: Comparison of accuracy between MDG and PMDG settings on the OfficeHome dataset under equal training data conditions. The x-axis shows the total number of training samples, while the y-axis shows the accuracy. Blue and orange lines represent MDG and PMDG settings, respectively. Each subplot corresponds to a different test domain: Clipart (top-left), Real-World (bottom-left), Product (top-right), and Art (bottom-right).

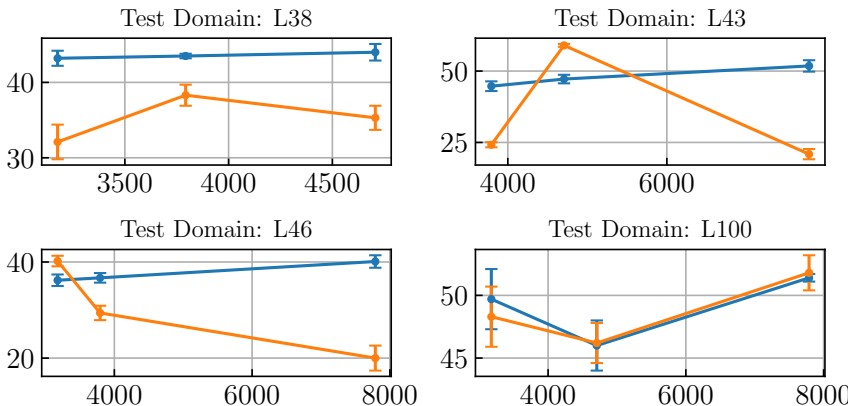

Figure 8: Comparison of accuracy between MDG and PMDG settings on the TerraIncognita dataset under equal training data conditions. The x-axis shows the total number of training samples, while the y-axis shows the accuracy. Blue and orange lines represent MDG and PMDG settings, respectively. Each subplot corresponds to a different test domain: L38 (top-left), L46 (bottom-left), L43 (top-right), and L100 (bottom-right).

## D    ADDITIONAL VISUALIZATION RESULTS

We present more visualization results in Figure 9, including a comprehensive comparison of various data augmentation and transformation techniques. The figure shows the visual effects of different approaches: No Data Augmentation as a baseline, Default Data Augmentation as a standard practice on DomainBed, and ten different transformation techniques including AugMix, CartoonGAN, CutMix, Edge Detection, IPMix, MixUp, RandAugment, RandConv, Style Transfer, and TrivialAugment. These visualizations provide insights into how each technique modifies the input images, offering a clearer understanding of their distinctive characteristics and potential impact on model training.

## E    THE USE OF LARGE LANGUAGE MODELS

We used LLMs to assist with the translation of technical content and to improve the clarity and readability of written sections. This included refining grammatical structures and enhancing sentence flow.

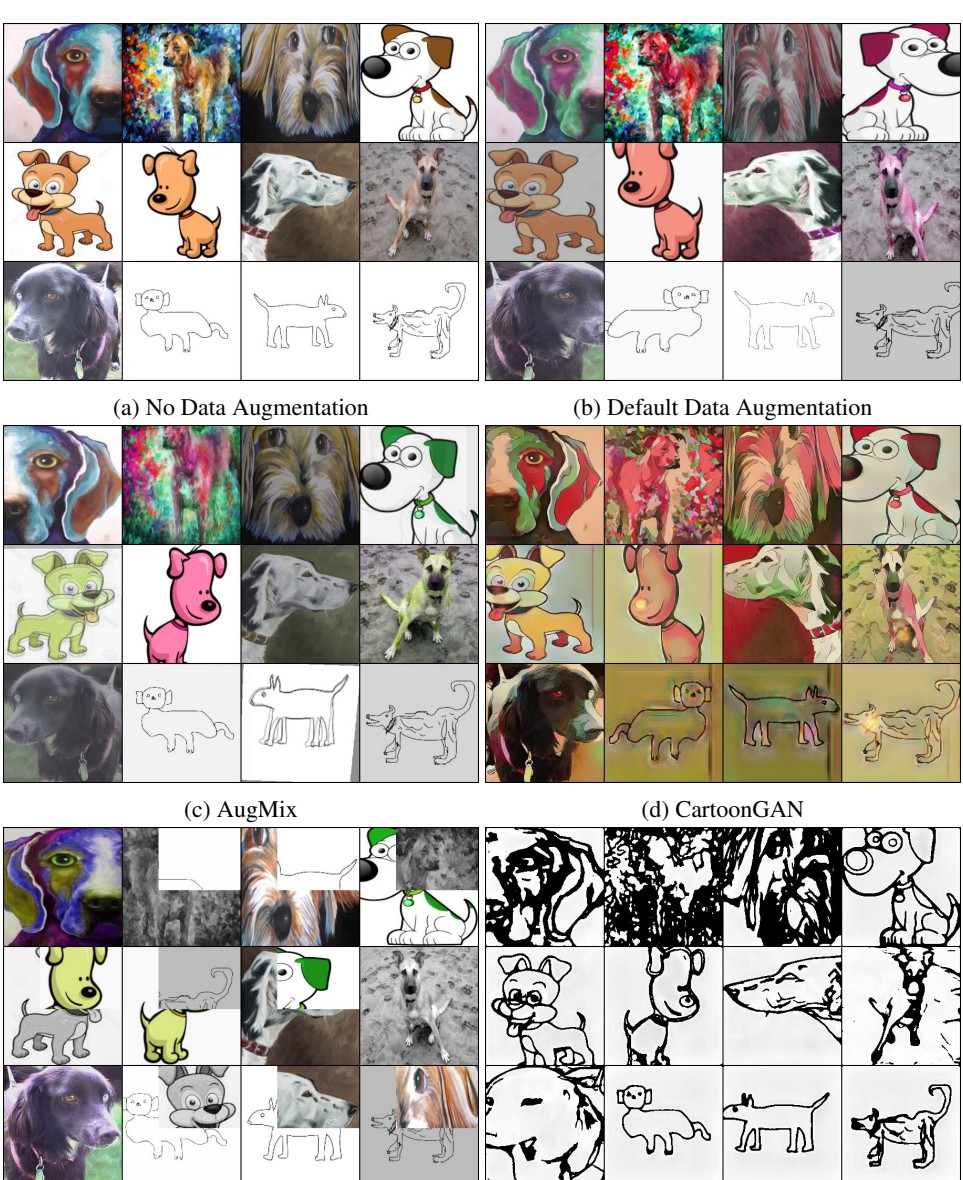



(a) No Data Augmentation       (b) Default Data Augmentation

(c) AugMix       (d) CartoonGAN

(e) CutMix       (f) Edge Detection



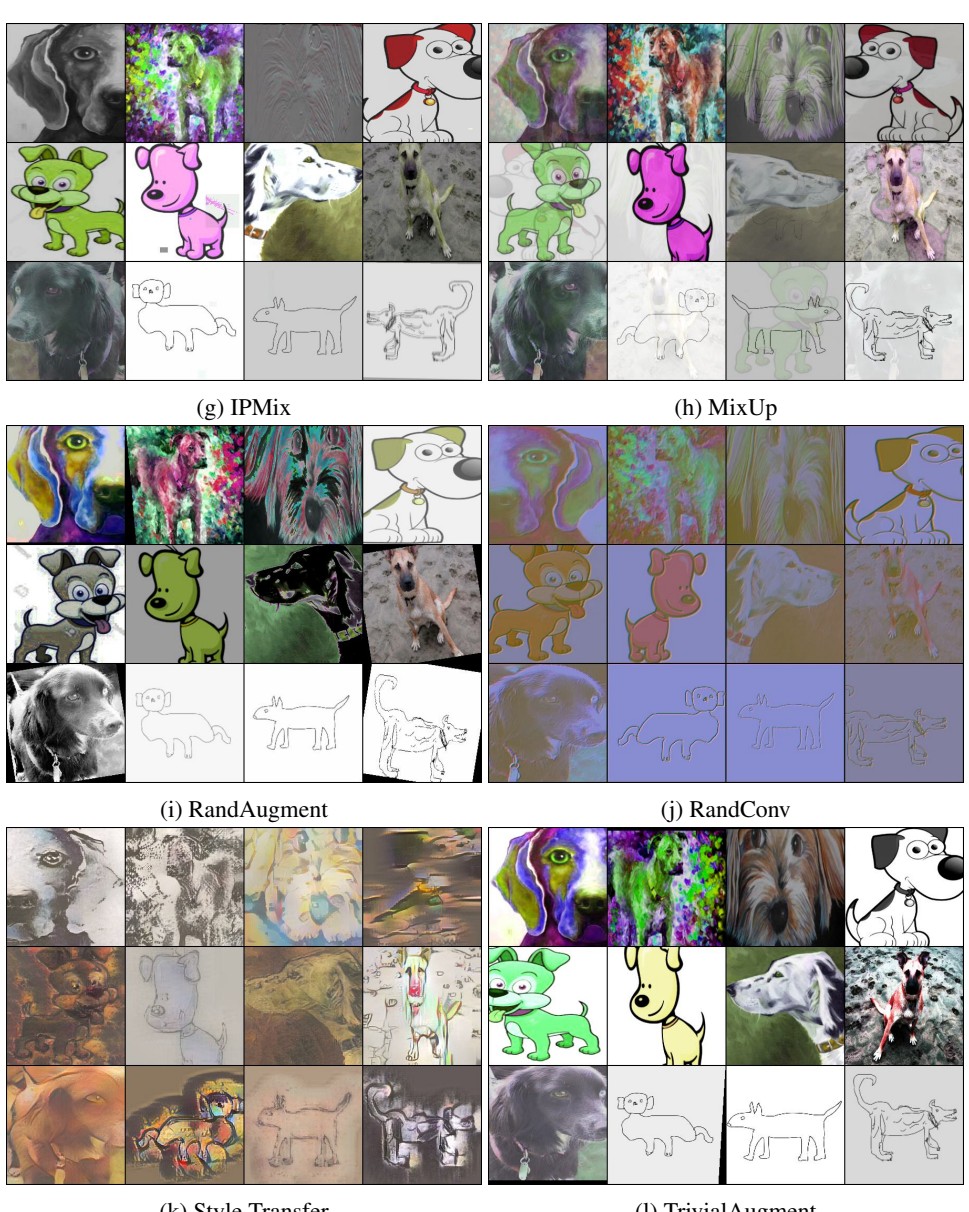

(g) IPMix

(h) MixUp

(i) RandAugment

(j) RandConv

(k) Style Transfer

(l) TrivialAugment

Figure 9: Comparison of Various Image Transformation Techniques. Default Data Augmentation refers to the standard data augmentation settings, while No Data Augmentation indicates the case where no data augmentation was applied.

