# OpenReview forum: "Pseudo Multi-Source Domain Generalization: Bridging the Gap Between Single and Multi-Source Domain Generalization"
_ICLR.cc/2026/Conference — ICLR 2026 Conference Withdrawn Submission_

### Official Review · Reviewer_n7Db · 2025-10-14

**Soundness:** 3
**Presentation:** 3
**Contribution:** 1
**Rating:** 0
**Confidence:** 5

**Summary:**

The paper proposes Pseudo Multi-Source Domain Generalization (PMDG): generate multiple “pseudo-domains” from a single source by applying style transfer (AdaIN, CartoonGAN, edge maps) and data augmentations (MixUp, CutMix, RandAugment, AugMix, IPMix, RandConv, etc.), then plug any multi-source DG (MDG) algorithm from DomainBed to train on these transformed splits. The authors also release PseudoDomainBed, a DomainBed variant adapted to single-source via pseudo-domains. They report that certain augmentation sets (notably repeated IPMix) with specific MDG algorithms (e.g., SD/RIDG) outperform single-source DG baselines; they also claim a positive correlation between MDG and PMDG rankings and occasional parity with true multi-source DG when data are abundant.

**Strengths:**

1. The work reuses DomainBed protocols, specifies transformation levels (dataset vs. minibatch) and interfaces, and turns off validation-time augmentation to avoid distortion.

2. Heatmaps across many MDG algorithms × transforms, comparisons of pseudo-domain combos, ViT vs. ResNet, and a data-efficiency curve (PMDG vs. MDG) are useful empirical references.

3. Authors explicitly note the shaky assumption that weak transforms are distinct domains and suggest measuring distribution distances

**Weaknesses:**

1. Core idea is largely a repackaging of well-trodden “domain expansion via augmentation.” Treating strong augmentations as different domains and training with MDG losses is conceptually aligned with years of SDG work on synthetic domain generation and adversarial/heuristic augmentations (e.g., MixStyle/RandConv/IPMix/OT/adversarial DA). PMDG’s “bridge” is effectively: apply known style/augmentation transforms, group by transform, and run existing MDG algorithms. This is incremental glue code rather than a new algorithmic principle.

2. Benchmarking contribution is a modest DomainBed fork. PseudoDomainBed mainly standardizes how augmentations are applied at dataset/minibatch levels; otherwise, it inherits MDG methods unmodified. The paper emphasizes breadth of runs more than new methodology.


3. No principled treatment of what constitutes a “domain.” The framework lacks a criterion to verify domain shift magnitude or diversity across pseudo-domains; intensity schedules, distribution overlap, and coverage are unaddressed (acknowledged in the limitation), leaving the method heuristic.

4. Lack of theoretical analysis.

**Questions:**

1. What is the substantive novelty beyond “augmentation-as-domains”?
Please delineate what PMDG adds algorithmically beyond grouping known transforms (style transfer, IPMix, RandConv, etc.) into K pseudo-domains and training with an off-the-shelf MDG loss. A precise contrast to “domain expansion” SDG prior art would help.


2. Can you justify that these pseudo-domains are truly “domains”?
Add quantitative evidence (e.g., FID/LPIPS, covariance shift, spectral stats, Wasserstein distances) showing that (i) each pseudo-domain differs from the source and (ii) pseudo-domains are mutually distinct. Tie these distances to generalization gains (correlation or causal ablation). Your discussion mentions this gap; closing it would strengthen the claim.

3. Automatic or principled transform selection.
Right now K transforms are chosen empirically. Could you propose an automatic selection/scheduling criterion and show it helps?

+See weaknesses above.

---

### Official Review · Reviewer_W1uA · 2025-10-26

**Soundness:** 3
**Presentation:** 2
**Contribution:** 2
**Rating:** 4
**Confidence:** 4

**Summary:**

This paper proposes to generate multiple pseudo-domains from a single source domain using style transfer and data augmentation techniques, enabling the application of sophisticated MDG algorithms. The authors introduce PseudoDomainBed, a modified DomainBed benchmark for the SDG setting, and conduct extensive experiments showing that PMDG can match or exceed traditional MDG approaches with sufficient data while outperforming existing SDG baselines. The key insight is that the quality of pseudo-domain generation may be more important than developing new learning algorithms.

**Strengths:**

1. The paper addresses a genuine practical limitation of MDG methods by proposing a straightforward and implementable solution that bridges SDG and MDG research directions.

2. The extensive experiments across multiple datasets, architectures, and MDG algorithms provide thorough validation of the PMDG framework and reveal useful insights about algorithm-domain shift compatibility.

3. The introduction of PseudoDomainBed with publicly available code facilitates reproducibility and future research.

**Weaknesses:**

1. The core idea of treating augmented data as distinct domains is relatively incremental, essentially applying existing data augmentation techniques and training with existing MDG algorithms. The paper lacks significant methodological innovation beyond the combination of known techniques.

2. The critical assumption that all transformed data represents distinct domains with distributions different from the source is not empirically or theoretically validated, particularly for weakly-transformed samples that may remain very similar to the original distribution.

3. The effectiveness of pseudo-domain combinations varies dramatically across datasets (e.g., style-based transformations excel on PACS but fail on TerraIncognita), and PMDG sometimes significantly underperforms MDG, raising questions about when practitioners should actually use this approach versus collecting real multi-domain data.

4. The paper does not quantitatively assess whether generated pseudo-domains actually exhibit sufficient distributional diversity or how to measure domain shift magnitude, making it unclear which transformation combinations should be preferred for specific application scenarios.

**Questions:**

Given the strong dataset-dependency of optimal transformation combination, how can practitioners determine which transformations to use for new domain generalization problems?

---

### Official Review · Reviewer_QdUp · 2025-10-27

**Soundness:** 3
**Presentation:** 3
**Contribution:** 2
**Rating:** 2
**Confidence:** 5

**Summary:**

The paper proposes the Pseudo Multi-source Domain Generalization (PMDG) framework for single-source domain generalization. PMDG generates multiple pseudo-domains from a single-source domain through style transfer and data augmentation techniques and use Multi-source Domain Generalization algorithms on the synthetic multi-domain dataset.  The experiment shows the effectiveness of proposed approach in domain generalization benchmark.

**Strengths:**

1. The proposed method shows the effectiveness of the PMDG on domain generalization benchmark.
2. The readability of this paper is good, and the structure is clear.

**Weaknesses:**

1. Single source domain generation methods often utilize data augmentation or generation strategy to expand the source data distribution. The difference between the proposed method and the existing data augmentation-based or domain expansion-based method is not clear.
2. Missing results compared with SOTA single source domain generalization methods on the benchmarks.
3.Some abbreviations are not clearly defined, such as the ST,ED,CG, and IM in the section 6.2. The expression "Org+IM+IM" in the second sentence of the section 6.2 includes two instances of "IM", the meaning of which is unclear.

**Questions:**

1. How to determine the kind of strategy used for Pseudo-domain generation?
2. Why is there a significant performance gap between MDG and PMDG when the test domain is G? (The results in Fig. 5)

---

### Official Review · Reviewer_jgeF · 2025-11-01

**Soundness:** 2
**Presentation:** 2
**Contribution:** 1
**Rating:** 2
**Confidence:** 4

**Summary:**

This paper a framework that generates pseudo multi-domain datasets for OOD generalization. The method generates multiple pseudo-domains from a single source domain using style transfer and data augmentation. Then it trains models using existing multi-source domain generalization (MDG) algorithms. The paper introduces PseudoDomainBed, a modified DomainBed benchmark for single-source settings. Experiments show PMDG can match or exceed MDG performance with sufficient data.

**Strengths:**

1. Bridging single-source and multi-source domain generalization addresses real needs. Collecting multi-domain datasets is expensive.

2. PseudoDomainBed provides standardized evaluation. Modified DomainBed for single-source setting with public code.

**Weaknesses:**

1. Using data augmentation as pseudo-domains is trivial. This is standard practice in domain generalization. Many existing SDG methods already do this. The paper just applies existing augmentations with existing MDG algorithms.

2. The core idea (treat augmentations as domains) is obvious. No new methods, no new algorithms, no new techniques. Just combining existing pieces.

3. The paper frames augmentation as "pseudo-domain generation" but this is just renaming existing techniques. Single-domain methods already use augmentation heavily.

4. Treats all augmentations as separate domains without validation. No evidence that augmented data represents real domain shifts. Likely just noise in many cases.

**Questions:**

See the Weakness

---

### Note · Authors · 2025-11-14

I have read and agree with the venue's withdrawal policy on behalf of myself and my co-authors.